# Unusual Clinical Manifestations in a Mexican Patient with Sanfilippo B Syndrome

**DOI:** 10.3390/diagnostics12051268

**Published:** 2022-05-19

**Authors:** Liliana Fernández-Hernández, Miriam Erandi Reyna-Fabián, Miguel Angel Alcántara-Ortigoza, Carmen Aláez-Verson, Luis L. Flores-Lagunes, Karol Carrillo-Sánchez, Ariadna González-del Angel

**Affiliations:** 1Laboratorio de Biología Molecular, Subdirección de Investigación Médica, Instituto Nacional de Pediatría, Insurgentes Sur 3700-C, Insurgentes Cuicuilco, Coyoacán, Mexico City CP 04530, Mexico; dralilianafernandez@gmail.com (L.F.-H.); erandif@yahoo.com (M.E.R.-F.); malcantaraortigoza@gmail.com (M.A.A.-O.); 2Laboratorio de Diagnóstico Genómico, Instituto Nacional de Medicina Genómica (INMEGEN), Periférico Sur 4809, Arenal Tepepan, Tlalpan, Mexico City CP 14610, Mexico; calaez@inmegen.gob.mx (C.A.-V.); lflores@inmegen.gob.mx (L.L.F.-L.); kcarrillo@inmegen.gob.mx (K.C.-S.)

**Keywords:** Sanfilippo B syndrome, MPS IIIB, unusual manifestations, growth arrest, sexual development, clinical exome sequencing

## Abstract

We present an unusual Mexican patient affected with mucopolysaccharidosis type IIIB (MPS IIIB; also called Sanfilippo B syndrome, MIM #252920) bearing clinical features that have not previously been described for MPS IIIB (growth arrest, hypogonadotropic hypogonadism, and congenital heart disease). Chromosomal microarray analysis was useful in identifying runs of homozygosity at 17q11.1–q21.33 and supporting the diagnosis of an underlying autosomal recessive condition. Sanger sequencing of *NAGLU* (17q21.2, MIM*609701) allowed us to identify a pathogenic homozygous p.(Arg234Cys) genotype. This *NAGLU* allele could be related to that previously described in an Iberian MPS IIIB founder haplotype; results from the polymorphic marker D17S800 and rs2071046 led us to hypothesize that it may have been introduced to Mexico through the Spanish settlement. The analysis of a clinical exome sequencing ruled out other monogenic etiologies for the previously undescribed clinical MPS IIIB manifestations. Our findings contribute to further delineating the MPS IIIB phenotype and suggest possible phenotype–genotype correlations.

## 1. Introduction

Mucopolysaccharidosis type IIIB (MPS IIIB; also called Sanfilippo syndrome B, MIM #252920) is an autosomal recessive inherited lysosomal storage disease caused by N-acetyl-alpha-glucosaminidase (NAGLU; E.C. 3.2.1.50) deficiency. Its estimated prevalence is 0.3 to 4.1 per 100,000 newborns [1]; however, this varies by region, with different rates reported for the Netherlands (1.89), Germany (1.57), France (0.68), the United Kingdom (1.21), and northern Portugal (0.84) [2]. The prevalence in Mexico is unknown. MPS IIIB is the most frequent MPS type in southeast Europe, where it reflects known migration patterns, mainly in Italy and Greece [3].

MPS IIIB is clinically characterized by severe degeneration of the central nervous system (CNS) with mild somatic disease. The symptoms frequently arise between 2 and 6 years of age and include developmental delay (85%), aggressive and hyperactive behavior (65%) [1], sleep disturbance (80–90%), and seizures. Affected individuals also show progressive loss of skills and the appearance of gait disorders and pyramidal signs. Disease progression eventually leads to death, generally early in the third decade of life [2].

MPS IIIB patients present a normal or slightly below average height and mild musculoskeletal abnormalities. In the context of pubertal development, hormonal alteration has not been reported in these patients. In a small proportion of patients, cardiac abnormalities, such as mild valvular leakage, ventricular hypertrophy, and atrial fibrillation, have been reported at clinical followup [4].

The diagnosis of MPS IIIB should include measurement of NAGLU enzymatic activity in blood serum, leukocytes, or fibroblasts [2]. Characterization of pathogenic variants in the *NAGLU* gene (17q21.2, MIM*609701) offers information that can be used to make a definitive MPS IIIB diagnosis and possibly a genotype–phenotype correlation. To date, about 171 pathogenic variants have been reported in the HGMD database (http://www.hgmd.cf.ac.uk/ac/gene.php?gene=NAGLU, accessed on 22 February 2022; most of them are missense and nonsense types (70%)).

Although there is no curative treatment, an early diagnosis allows access to new therapies. To date, there is only one clinical trial, with 15 MPS IIIB patients with promising results in terms of efficacy and improvement in quality of life. (Biological: rAAV9.CMV.hNAGLU) (https://clinicaltrials.gov/ct2/results?cond=&term=MPSIIIB&cntry=&state=&city=&dist=&Search=Search, accessed on 14 April 2022).

Here, we present the first Mexican MPS IIIB patient homozygous for a *NAGLU* pathogenic variant. Interestingly, this patient shows an important growth arrest, hypogonadotropic hypogonadism, and congenital heart disease, none of which has been previously reported as a clinical manifestation of this entity.

## 2. Case Report

We describe a 21-year-old boy. He is the first child of healthy and endogamic parents, born in a small community of 235 inhabitants in Nueva Santa Lucía, Chapantongo, Hidalgo, Mexico. At the time of his gestation, his mother was 18 years old and his father was 22 years old. The patient has three healthy siblings. Family and gestational history were unremarkable. Perinatal attention was performed at 40 weeks of gestation after spontaneous vaginal delivery. Birth weight was 2.250 kg (Z = −3.12), length was 51 cm (Z = −0.65), APGAR was 8/9, and the occipital-frontal circumference was not recorded. The parents did not report any complications at birth. He was referred to our institution at 9 years of age, because he exhibited delayed speech and language development (HP:0000750) and the loss of some acquired skills (HP:0002376), mainly sphincter control and follow-up instructions, after exhibiting normal development up to 4 years of age. On physical examination, his weight was 19.500 kg (Z = −3) and his height was 106 cm (Z = −4); his midparental height is 166.5 cm (Z = −1.7). His head circumference was 48.4 cm (Z = −3) and he exhibited a coarse face (HP:0000280), left palpebral ptosis (HP:0007687) (Figure 1A), and bilateral limitation to elbow extension (HP:0001377). An elevated concentration of uronic acid (78.1 µg; normal range < 15 µg) was detected in urine. No corneal clouding was found on ophthalmologic evaluation, so we assessed for mucopolysaccharidosis II (MPS II) by quantifying the enzymatic activity of iduronate-2-sulfatase in a dried blood spot and performing a molecular study of the *IDS* gene. Our results ruled out MPS II. The patient’s arylsulfatase B was in the normal range (measured as an internal enzymatic test control), which ruled out MPS VI. The abdominal ultrasound and X-ray evaluation were normal and did not show evidence of visceromegaly or dysostosis multiplex. Unfortunately, the enzymatic activity assays needed to confirm a MPS IIIA or MPS IIIB diagnosis were not available in our country.

We detected an 8-mm *ostium secundum* type atrial septal defect (ORPHA:99103) that lacked hemodynamic repercussion and did not require medical or surgical management. At 10 years of age, the patient’s growth halted (HP:0001510) (Figure 1B); he exhibited a normal growth hormone level (1.4 ng/mL, normal range 0.1–10 ng/mL) with no evidence of anemia, infection, or renal disorder.

### Molecular Characterization

Given the patients’ growth arrest and congenital heart disease, we started an integral diagnostic approach. G-band karyotyping of peripheral lymphocytes revealed a karyotype of 46, XY [15] at a 450 GTG-band resolution. Metabolic screening of a dried blood sample, including amino-acid and acylcarnitine profiling, yielded results that were within normal limits. Brain magnetic resonance imaging (MRI) demonstrated mild ventriculomegaly (HP:0002119) and brain atrophy (HP:0012444). During a clinical followup at 14 years of age, the patient presented tonic–clonic seizures (HP:0002069) that were well controlled with oxcarbazepine; to date, he has presented an episode approximately every 2 months. At the age of 14 years, we also noticed an absence of pubertal development (HP:0008197). Hypogonadism (HP:0000135) was identified through hormonal profiling, which revealed the following values: LH < 0.10 mUI/mL (normal pubertal range 0.4–7.0 mUI/mL), FSH 0.9 mUI/mL (normal pubertal range 1.0–12.8 mUI/mL), and testosterone < 20 ng/dL (normal pubertal range 100–1000 mUI/mL). Considering the patient’s growth arrest (at 18 years of age, his weight was 19.200 kg (Z = −14), his height was 112 cm (Z = −9), and his head circumference was 49.2 cm (Z = −3)), the absence of puberty, and the presence of congenital heart disease, we decided to carry out a chromosomal microarray analysis (CMA; CytoScanTM High Density Microarray; Affymetrix, Inc., Santa Clara, CA, USA). The CMA revealed three regions of runs of homozygosity (ROH): 1q31.1–q32.1: 15.30 Mb (186,749,232–202,054,905 GRCh37) involving 53 protein-coding genes, 6p12.3–p12.1: 5.69 Mb (48,378,643–54,071,413 GRCh37) involving 37 protein-coding genes, and 17q11.1-q21.33: 22.78 Mb (25,309,336–48,094,611 GRCh37) involving 448 protein-coding genes, the latter (Figure 2A) including the *NAGLU* gene. Based on this result, the clinical manifestations, and our initial diagnostic suspicion, we implemented a molecular analysis involving PCR amplification followed by direct Sanger sequencing of all six coding exons of *NAGLU*. This analysis revealed the NM_000263.3(*NAGLU*):c.[700C>T];[700C>T] or p.[Arg234Cys];[Arg234Cys] genotype in homozygous state (Figure 2B). The heterozygous state was confirmed in both parents and in the youngest sister, while the other two siblings were not available for molecular testing.

The complete *NAGLU* sequencing performed in our patient revealed additional variants. The patient was homozygous for the G/G genotype at the “IVS2+50G>C” (rs2071046:G>C) site previously described in three unrelated homozygous p.(Arg234Cys) MPS IIIB Portuguese and Spanish patients [6]. Additionally, three benign variants were identified in homozygous state for the minor alleles: NM_000263.4(*NAGLU*):c.423T>C (rs659497), NM_000263.4(*NAGLU*):c.1021+120C>T (rs630539) and NM_000263.4(*NAGLU*):c.2209C>G (rs86312). This correlates with the ROH identified for the 17q11.1-q21.33 region. To determine if the p.(Arg234Cys) pathogenic allele in our patient could share a common origin with those of the previously reported Portuguese and Spanish patients [6], we analyzed the polymorphic microsatellite marker D17S800 (located inside the ROH region at 17q11.1-q21.33) by Sanger sequencing and fluorescent fragment analysis by capillary electrophoresis (assay conditions are available upon request). These analyses revealed a heterozygous 174/176 genotype.

To elucidate whether the observed growth arrest, lack of puberty, and congenital heart disease that had not been previously described in MPS IIIB were caused by pathogenic variants in other genes, we performed a clinical exome sequencing (CES). Library preparation was performed using the reagents provided in the Clinical Exome sequencing panel kit, version 2 (Sophia Genetics SA, Saint Sulpice, Switzerland), according to the manufacturer’s protocol. Sequencing was performed on a NextSeq Instrument (Illumina, San Diego, CA, USA). Sequencing data analysis and variant annotation were performed with the Sophia DDM^®^ software version 5.10.12 (Sophia Genetics SA), and FASTQ files were aligned to the GRCh37/hg19 human reference genome. Virtual filters were constructed for short stature and hypogonadism phenotypes according to the HPO database. The average read depth was 52X, with 98.22% of the target regions covered by at least 25x. The CES results revealed two clinically relevant variants in heterozygous state in the *TFR2* (MIM*604720) gene, which is responsible for hemochromatosis type 3 (HFE3, MIM#604250): NM_003227.4:c.1629del or p.(Gln544Argfs*18) [rs749211542] and NM_003227.4:c.283G>A or p.(Gly95Arg) [rs746436648]. These changes were classified as pathogenic and of unknown significance (VUS), respectively. As HFE3 has autosomal recessive inheritance, we assessed the *TFR2* genotype in the parents by Sanger sequencing. This analysis revealed that both variants were found in a single maternal haplotype *(-cis).* Finally, we carried out a plasma blood transferrin study in the index case. The results ruled out the HFE3 phenotype. No other clinically relevant variants that could be related to those three MPS IIIB undescribed manifestations were identified in the CES study.

## 3. Discussion

MPS IIIB is a rare metabolic disorder with a discouraging prognosis due to the lack of a curative therapy. Diagnosis rests on clinical suspicion, enzymatic measurements, and molecular analysis. Unfortunately, the latter two are not systematically performed in Mexico. To our knowledge, this is the first MPS IIIB patient reported in Mexico with a confirmatory *NAGLU* genotype. Interestingly, our patient shared many of the reported manifestations of MPS III and further presented three clinical signs that have not previously been described in other patients: growth arrest, lack of puberty, and congenital heart disease.

Kamp et al. reported a series of 73 MPS III patients (23 with MPS IIIB); among them, growth was affected in approximately half of the patients older than 12 years, but the final height was usually within normal percentiles [5]. In contrast, the MPS IIIB patient studied herein presented a growth arrest that started at 9 years of age and reached a final height with a Z score = −9. Other known factors related to short stature were discarded, as he presented normal routine laboratories studies and a normal growth hormone level of 1.4 ng/mL (normal range 0.1–10 ng/mL). Other common monogenic causes associated with this trait were discarded by our CES analysis. However, as height demonstrates a polygenic etiology, many interactions could be responsible for the growth arrest observed in this patient.

A lack of pubertal development has not previously been reported in an MPS IIIB patient, and adequate sexual development was noted in a case series of patients older than 64 years [7]. However, our patient showed clinical and biochemical signs indicative of hypogonadotropic hypogonadism. Many genes have been implicated in diseases that course with this trait, but some of them were discarded as causative by our CES analysis. Nevertheless, as this manifestation is not consistently found in MPS IIIB patients, it also may reflect other gene variants and/or oligogenic interactions that should be further assessed.

Cardiac involvement in MPS III is less common and milder that that seen in other types of MPS; it is frequently characterized by valvular disease and cardiac hypertrophy due to disease progression. However, our patient presented an atrial septal defect (*ostium secundum* type), which has not previously been reported in MPS IIIB. Lin et al. described the follow-up of 26 Taiwanese MPS III patients (20 with MPS IIIB); among them, the most prevalent cardiac abnormality was mitral regurgitation (31%) followed by aortic regurgitation (19%), and no studied patient had a cardiac septal defect [8]. Isolated *ostium secundum* defects account for 7% of overall congenital heart disease and represent the most common form of atrial septal defect [9]. Given its estimated frequency in the general population, this congenital heart defect could be coincidental in our patient. However, it could also be an unusual MPS IIIB manifestation.

The identified missense *NAGLU* variant, c.700C>T or p.(Arg234Cys), was previously reported as pathogenic and was first described in the homozygous state in one Spanish patient with a severe MPS IIIB phenotype (NAGLU enzyme activity of <0.1 nmol/h/mg protein, reference range 1.2–4.6 nmol/h/mg protein) [10]. A second report documented a compound heterozygous p.[Arg234Cys]; p.[Trp361Arg] Chinese patient who also showed severe manifestations (coarse facial features, intellectual disability, hepatomegaly, and undetectable NAGLU activity) [11]. However, neither patient manifested the unusual clinical features described herein. The p.(Arg234Cys) variant comprises 32% of the pathogenic *NAGLU* alleles found in Spanish and Portuguese patients; it originated from the Iberian Peninsula, and a founder haplotype has been described [6].

Given that our patient came from a small inbred community, the ROH at the 17q11.1–q21.33 region, a homozygous *NAGLU* haplotype comprising the minor alleles of the rs659497, rs86312, and rs630539 sites, and G/G homozygosity at the “IVS2+50G>C” (rs2071046:G>C) site, a founder effect for p.(Arg234Cys) in this community should not be discarded. Although Mangas et al. did not report genotypes for rs659497, rs86312, or rs630539, their report of G/G homozygosity for “IVS2+50G>C” (rs2071046:G>C) led us to hypothesize that the herein identified *NAGLU* haplotype could be related to the proposed Iberian p.(Arg234Cys) founder haplotype. Notably, the G allele of rs2071046 was always observed in complete linkage disequilibrium with p.(Arg234Cys) in Portuguese and Spanish patients [6], leading us to postulate that it may have been introduced to Mexico through the Spanish settlement. The 174 allele identified for marker DS17S800 in our patient agrees with this hypothesis, while the 176 allele, which is unrelated to the Iberian *NAGLU* p.(Arg234Cys) founder haplotype [5], could be attributed to a recent mutational event in light of the high expected mutation rate for dinucleotide-type short tandem repeats. Indeed, unlike the G allele of rs2071046, the 174 allele of D17S800 could be unequivocally linked with p.(Arg234Cys) in only 10 of the 19 previously studied patients [6]. Additional genotyping for D17S1801 in our patient and/or for the rs659497, rs86312, and rs630539 markers in Iberian patients could help support a common origin between the herein identified p.(Arg234Cys) *NAGLU* haplotype and that reported to be of Iberian origin.

There is little information available regarding the number of MPS IIIB patients in Mexico. A local online register database (http://www.redsanfilippo.org, accessed on 22 February 2022), contains only 15 MPS IIIB patients (personal communication). According to the INMEGEN database, the p.(Arg234Cys) *NAGLU* variant was identified in one allele of another Mexican MPS patient, suggesting that this mutation may be frequent in this country. The potential underdiagnosis of MPS IIIB in Mexico could reflect a combination of various factors, including the lack of neonatal screening, the absence of enzymatic replacement therapy, and limitations in performing routine molecular and biochemical confirmatory studies. Such factors can cause a delay in diagnosis, as illustrated for our patient.

## 4. Conclusions

Here, we reported the first Mexican MPS IIIB patient bearing a homozygous p.(Arg234Cys) *NAGLU* genotype. This patient displayed the previously undescribed manifestations of growth arrest, hypogonadotropic hypogonadism, and congenital heart disease. A monogenic origin for these traits was discarded by CES analysis. Further study is warranted to determine if this patient’s genotype is related to a previously proposed founder Iberian MPS IIIB-causing haplotype. Despite the diagnostic delay in our patient, which was attributed to the patient’s atypical clinical manifestations and lack of access to enzymatic and molecular MPS IIIB confirmatory analyses in Mexico, the present case illustrates the usefulness of CMA in identifying ROH to support the diagnosis of an underlying autosomal recessive condition in a patient whose clinical phenotype is atypical.

## Figures and Tables

**Figure 1 diagnostics-12-01268-f001:**
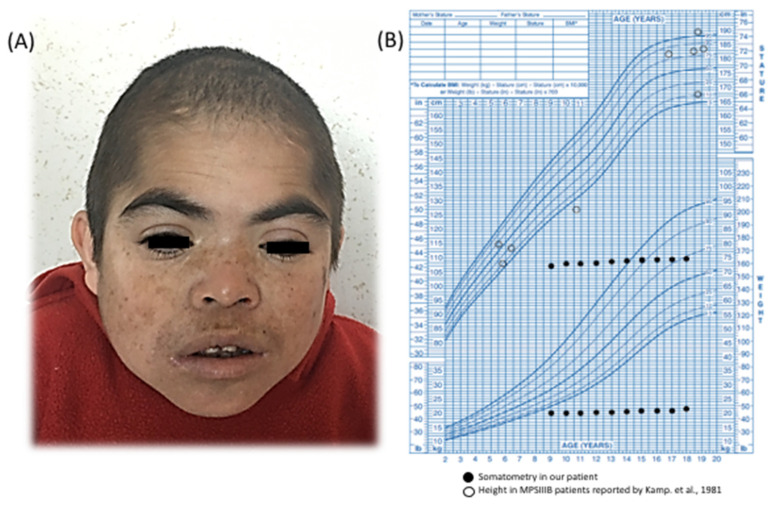
(**A**) Photograph of the patient at 18 years of age, showing bifrontal narrowing, thick eyebrows, left palpebral ptosis, thick lips, and a coarse face. (**B**) Weight and size chart of the general population showing the arrest of both parameters in the present patient and those previously reported by Kamp et al. [5].

**Figure 2 diagnostics-12-01268-f002:**
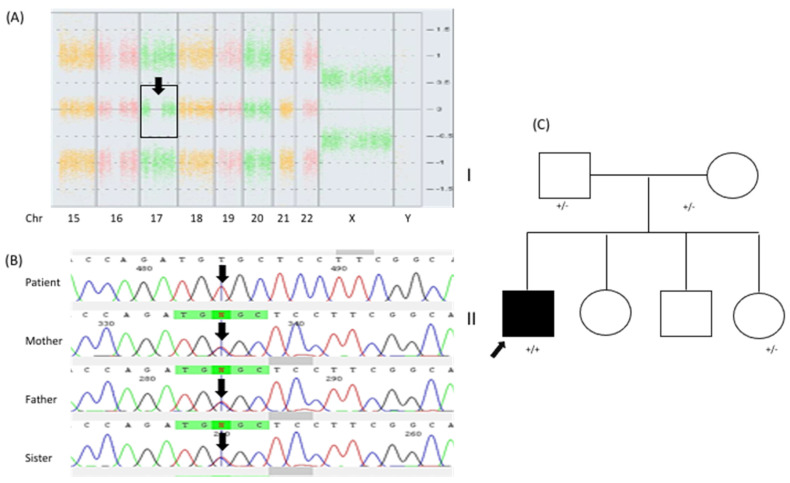
Results of the molecular studies. (**A**) CMA plot showing the 22.78−Mb ROH at 17q11.1-q21.33 (Chr17: 25,309,336-8,094,611 GRCh37). (**B**) Partial electropherogram of exon 3 of the *NAGLU* gene, indicating (black arrow) the homozygous pathogenic variant NM_000263.3:c.700C>T or p.(Arg234Cys) in index case II.2, and showing that both parents I.1 and I.2 and the youngest sister II.3 were heterozygous for the variant. (**C**) Two-generation patient’s pedigree.

## Data Availability

Not applicable.

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
