# Peer review of "Unusual Clinical Manifestations in a Mexican Patient with Sanfilippo B Syndrome"

_diagnostics, 2022, doi:10.3390/diagnostics12051268_

Round 1
Reviewer 1 Report
The manuscript entitled "Unusual clinical manifestations in a Mexican patient with San- 2 filippo B syndrome" is a well-written, interesting article describing a Mexican patient with the most common type of MPS in southeast Europe. But i have some questions and suggestions for the clarity of the molecular genetic analysis results.
- If we talk about "loss of heterozygosity" we generally indicate that one of the alternating allele of the parental genome is not contributing the current content. In the presented case, both of the parents were heterozygous carriers explaining the homozygosity of the patient. How do authors conclude that the patient has only one copy of the related region of the NAGLU gene? or the copies belong to one of the parents only. I would prefer to see more detailed information on the microarray results if possible. The microarray image given is not explanatory according to my opinion.
- Patient has ptosis in his left eye, I suggest this may be added to dysmorphic features explained in the text.
- For the sentence "This analysis revealed the homozygous pathogenic gen otype NM_000263.3(NAGLU):c.[700C>T];[700C>T] or p.[Arg234Cys];[Arg234Cys]" in the 118th line; I would not prefer to write homozygosity in this form. Only describing the situation is enough as"homozygous" or "heterozygous" . This kind of writing (with a /) for defining the mutations zygosity does not listed in HGCV guidelines as far as I know.
Author Response
We are grateful to the reviewers for their valuable comments regarding our manuscript and we have considered all of their suggestions. Our responses to each comment are listed below. Also, we marked the changes using the “Track
Changes” function in the main manuscript.
REVIEWER 1
The manuscript entitled "Unusual clinical manifestations in a Mexican patient with Sanfilippo B syndrome" is a well-written, interesting article describing a Mexican patient with the most common type of MPS in southeast Europe. But I have some questions and suggestions for the clarity of the molecular genetic analysis results.
1. If we talk about "loss of heterozygosity" we generally indicate that one of the alternating alleles of the parental genome is not contributing the current content. In the presented case, both of the parents were heterozygous carriers explaining the homozygosity of the patient. How do authors conclude that the patient has only one copy of the related region of the NAGLU gene? or the copies belong to one of the parents only. I would prefer to see more detailed information on the microarray results if possible. The microarray image given is not explanatory according to my opinion.
RESPONSE: Thanks a lot for your comment. We agree with the definition you mentioned about “loss of heterozygosity” and as you also referred, we confirmed that the parents of the patient were heterozygous for the pathogenic variant in NAGLU gene, so according to the array study and the molecular analysis of some SNPs, the patient has two copies of the related region of the NAGLU gene with the contribution of both parents. The chromosomal microarray (CMA) analysis revealed homozygosity at 17q11.1-q21.33 region. This feature in the literature are also known as Runs of Homozygosity (ROH, contiguous regions of the genome where an individual is homozygous across all sites) (Ceballos et al. BMC Genomics (2018) 19:106), so we consider more appropriate to replace the term of “loss of heterozygosity” with ROH along text. The ROH at 17q11.1-q21.33 region identified by CMA, which allows is indicated by an arrow in figure 2. This ROH allowed us to consider the diagnosis of Sanfilippo syndrome in the patient, with further confirmation by the Sanger sequencing of NAGLU gene.
2. Patient has ptosis in his left eye, I suggest this may be added to dysmorphic features explained in the text.
RESPONSE: We thank your valuable comment, we have added left palpebral ptosis to the clinical description of the patient.
Line 85: His head circumference was 48.4 cm (Z=-3) and he exhibited a coarse face (HP:0000280), left palpebral ptosis (HP:0007687) (Figure 1a) and bilateral limitation to elbow extension (HP:0001377).
Figure 1A: Figure 1. A) Photograph of the patient at 18 years of age, showing bifrontal narrowing, thick eyebrows, left palpebral ptosis, thick lips, and a coarse face.
3. For the sentence "This analysis revealed the homozygous pathogenic genotype NM_000263.3(NAGLU):c.[700C>T];[700C>T] or p.[Arg234Cys];[Arg234Cys]" in the 118th line; I would not prefer to write homozygosity in this form. Only describing the situation is enough as "homozygous" or "heterozygous". This kind of writing (with a /) for defining the mutations zygosity does not listed in HGCV guidelines as far as I know.
RESPONSE: We appreciate such a valuable suggestion, we have changed the text by exclusively describing the "homozygous" and/or "heterozygous" state.
Line 133-135: This analysis revealed the NM_000263.3(NAGLU):c.[700C>T];[700C>T] or p.[Arg234Cys];[Arg234Cys] genotype in homozygous state.
CHANGES BY THE AUTHOR
- We would like to change the term LOH (Loss of heterozygosity) for ROH (runs of homozygosity) throughout the manuscript, because it better describes the contiguous lengths of homozygous genotypes that are present in an individual due to parents transmitting identical haplotypes.
- We want to explain at the Ethics Statement that: Written informed consent was obtained from the patient’s parents for publication of this case report and any accompanying images. The present report is exempt from the ethical committee approval code because the ethics and research committee of the National Institute of Pediatrics do not review the diagnostic workup provided as part of the patient's medical assistance.
Reviewer 2 Report
Dear Authors, this is a well written paper. It is very important an early diagnosis in a patient, even if any specific treatment is available at the moment for MPS IIIB. The picture of the patient shows a very evocative face but the atypical clinical features of the child are well detailed and interesting.
We suggest to insert some informations about clinical trial of MPSIIIB.
Author Response
We are grateful to the reviewers for their valuable comments regarding our manuscript and we have considered all of their suggestions. Our responses to each comment are listed below. Also, we marked the changes using the “Track
Changes” function in the main manuscript.
COMMENTS TO THE AUTHOR
REVIEWER 2
Dear Authors, this is a well written paper. It is very important an early diagnosis in a patient, even if any specific treatment is available at the moment for MPS IIIB. The picture of the patient shows a very evocative face but the atypical clinical features of the child are well detailed and interesting.
We suggest to insert some information about clinical trial of MPSIIIB.
RESPONSE: We appreciate such a valuable suggestion, we have added a paragraph with the current clinical trials available for MPSIIIA
Line 57-63: “Although there is no curative treatment, an early diagnosis allows access to new therapies. To date there are 11 clinical trials: 3 recruiting patients (drug: ABO-102), 2 active (autologous CD34+ cells transduced with a lentiviral vector containing the human SGSH gene and LYS-SAF302) and 6 already completed utilizing: SOBI003, HGT-1410 and Recombinant human heparan N-sulfatase (rhHNS), all of them with promising results in accordance with the safety of the biological diversity used (https://clinicaltrials.gov/ct2/results?term=MPSIIIA&Search=Search)”.
Reviewer 3 Report
Fernández-Hernández and colleagues present a case report of a Mexican patient affected with mucopolysaccharidosis type IIIB bearing clinical features that have not previously been described for MPS IIIB (growth arrest, hypogonadotropic hypogonadism, and congenital heart disease). The reporting of individuals with this rare condition is of value to the literature. However, this manuscript needs a significant degree of editing (see below).
Materials and Methods
1.“Birth weight was 2.250 kg, length was 51 cm, APGAR was 8/9, and the occipital-frontal circumference was not recorded.” In Page 2, Line 68.
=> Could you tell me the Z-score of his birth weight and length?
2.“The patient’s arylsulfatase B was in the normal range (measured as an internal enzymatic test control), which ruled out MPS VI.” In Page 2, Line 80.
=> Did you consider to check the level of Iduronidase to rule out MPS I at that time?
3.” At 10 years of age, the patient’s growth halted (HP:0001510) (Figure 1b); he exhibited a normal growth hormone level (1.4 ng/ml, normal range 0.1-10 ng/ml) with no evidence of anemia, infection, or renal disorder.” In Page 2, Line 88.
=> Did you have clonidine test to rule out growth hormone deficiency?
Results
1.” Hypogonadism (HP:0000135) was identified through hormonal profiling, which revealed the following values: LH <0.10 mUI/mL (normal pubertal range 0.4-7.0 mUI/mL), FSH 0.9 mUI/mL (normal pubertal range 1.0-12.8 mUI/mL), and testosterone <20 ng/dL (normal pubertal range 100-1000 mUI/mL).” In Page 3, Line 106.
=> Did you have GnRH test for his delay puberty?
2.” The CMA revealed three regions of loss of heterozygosity (LOH): 1q31.1-q32.1, 6p12.3-p12.1, and 17q11.1-q21.33, the latter (Figure 2a) including the NAGLU gene.” In Page 3, Line 113.
=> Please tell us the length of LOH in 1q31.1-q32.1, 6p12.3-p12.1, and 17q11.1-q21.33. And how many OMIM genes in these LOHs.
3.” Additionally, four benign variants were homozygous for the minor alleles: rs659497:T>C, c.764+119C>T, rs86312:C>G, and rs630539:C>T.” In Page 4, Line 132.
=> rs659497:T>C should be NM_000263.3(NAGLU):c.423T>C (p.Ser141Ser). c.764+119C>T: I cannot find this variant in ClinVar.
rs86312:C>G should be NM_000263.4(NAGLU):c.2209C>G (p.Arg737Gly).
rs630539:C>T should be NM_000263.4(NAGLU):c.1021+120C>T
4.” This agrees with the LOH identified for the 17q11.1-q21.33 region.” In Page 4, Line 134.
=> Why these 4 benign variants agrees with the LOH in this patient?
Figure 2
1.This patient has 3 siblings, but figure 2B showed only 2 siblings.
2.Does his other relative have similar symptoms?
Reviewer 4 Report
This is a case report on a Mexican patient with Sanfilippo B syndrome who presents with unusual clinical manifestations.
In my opinion, the report adds on the phenotypic spectrum of Sanfilippo B syndrome associated with a specific NAGLU mutation, and thus it might be of interest specially to the community dealing with lysosomal storage disorders.
However, the current version would benefit from a few revisions, as I will next list:
- Organization of the entire manuscript. As a CASE REPORT, it does not make sense to present a “2. Material and methods” section containing a unique subsection numbered “2.1 Case report”. Omit the entire M &M section, and instead organise the manuscript going directly to the “Case Report”, subdividing it in subsections such as “Phenotypic presentation”, “Molecular characterization” and so on….
- Given the privacy concerns regarding the patient photo, it should be considered the possibly of de-identifying the face, by masking the eyes (and preferably also eyebrows).
- Lines 56-57; “…patient with a confirmatory NAGLU genotype”, Commonly, “confirmatory” is used to quality some kind of tests, not genotypes. I rather preferred something more conventional, such as “…patient who was homozygous for a NAGLU pathogenic mutation”.
- Lines 62-63; The sentence “He is the first child healthy, probably consanguineous (their grandparents share surnames) and endogamic parents,…” needs clarification. In this context, what is the difference between parental endogamy and parental consanguinity? Besides sharing the same surname, there is no more information on the parents?
- Line 72 “mainly sphincter control and following directions”. It is not clear what is being described. Rephrase.
- Figure 2 contains 3 images (A, B and C) but in the legend only two are mentioned: A and B. The description of B is incorrect.
- Line 120. Omit “obligate”.
- Line 158. Explain better how did you conclude that the 2 variants in TFR2 were transmitted by the maternal side. If the father was not carrier of any of the variations, and the mother was for both like the patient, then…..
- Line 210; “… and had probable endogamy…” Presuming that the subject in the sentence is the patient, it is better to say that “ and he is probably inbred”
- Lines 213-214. “….we believe it is reasonable to propose a founder effect for p.(Arg234Cys)”. I don’t understand the need to introduce here the concept of founder effect. It is enough to present the interpretation given soon after, whereby the authors postulate that the mutation was introduced into Mexico by Iberian people, likely during the Spanish settlement. In small endogamous communities, it is expected an increase in homozygosity, including for deleterious mutations.
- . Lines 235-236. To what extent the absence of enzymatic replacement therapy might contribute to the underdiagnosis of MPS IIIB in Mexico?
- The Discussion section should be shortened.
- I have a last question to the authors. In the regions of chromosome 1 and 6 revealing LOH, which genes are there? In the manuscript, a small comment on that would be welcome.
Round 2
Reviewer 3 Report
Thank you for providing your revised manuscript. This has now been checked by the Editor and the original reviewers. The reviewers have requested that you make some further changes. I have included these comments below:
Introduction
- “Although there is no curative treatment, an early diagnosis allows access to new therapies. To date there are 11 clinical trials: 3 recruiting patients (drug: ABO-102), 2 active (autologous CD34+ cells transduced with a lentiviral vector containing the human SGSH gene and LYS-SAF302) and 6 already completed utilizing: SOBI003, HGT-1410 and Recombinant human heparan N-sulfatase (rhHNS), all of them with promising results in 61 accordance with the safety of the biological diversity used (https://clinicaltrials.gov/ct2/results?term=MPSIIIA&Search=Search).” In Page 2, Line 57.
=> Could you tell us that these therapies are applied for which types of MPSIII(A/B/C/D) ?
Author Response
REVIEWER 3
- “Although there is no curative treatment, an early diagnosis allows access to new therapies. To date there are 11 clinical trials: 3 recruiting patients (drug: ABO-102), 2 active (autologous CD34+ cells transduced with a lentiviral vector containing the human SGSH gene and LYS-SAF302) and 6 already completed utilizing: SOBI003, HGT-1410 and Recombinant human heparan N-sulfatase (rhHNS), all of them with promising results in 61 accordance with the safety of the biological diversity used (https://clinicaltrials.gov/ct2/results?term=MPSIIIA&Search=Search).” In Page 2, Line 57.
=> Could you tell us that these therapies are applied for which types of MPSIII(A/B/C/D) ?
RESPONSE: We appreciate your observation. The therapies are for MPSIIIA and B. For MPSIIIB there are only one clinical trial, so, we have changed the sentence.
Line 59-63: “Although there is no curative treatment, an early diagnosis allows access to new therapies. To date, there is only one clinical trial, active, no recruiting patients, with 15 MPSIIIB patients with promising results in terms of efficacy and improvement in quality of life. (Biological: rAAV9.CMV.hNAGLU) https://clinicaltrials.gov/ct2/results?cond=&term=MPSIIIB&cntry=&state=&city=&dist=&Search=Search)”
Reviewer 4 Report
Thank you for considering my comments/suggestions.
Concerning my point on the founder effect, I didn't mean that a founder effect should be disregarded. My caution was because it is being describing a unique patient. So instead the phrase (in line 233) “….a founder effect for p.(Arg234Cys) should be discarded.” I suggest something like: “….a founder effect for p.(Arg234Cys) in this community should not be discarded.”
Author Response
REVIEWER 4
Concerning my point on the founder effect, I didn't mean that a founder effect should be disregarded. My caution was because it is being describing a unique patient. So instead the phrase (in line 233) “….a founder effect for p.(Arg234Cys) should be discarded.” I suggest something like: “….a founder effect for p.(Arg234Cys) in this community should not be discarded.”
RESPONSE: We appreciate your comment. We have changed the sentence.
Line 277-281: Given that our patient came from a small community and he is probably inbred, the ROH at the 17q11.1-q21.33 region, a homozygous NAGLU haplotype comprising the minor alleles of the rs659497, , rs86312, and rs630539 sites, and G/G homozygosity at the “IVS2+50G>C” (rs2071046:G>C) site, a founder effect for p.(Arg234Cys) in this community should not be discarded.